# Single cell tracing of Pomc neurons reveals recruitment of 'Ghost' subtypes with atypical identity in a mouse model of obesity

Stéphane Leon [1], Vincent Simon [1], Thomas H. Lee[1], Lukas Steuernagel [2], Samantha Clark[1], Nasim Biglari[2], Thierry Lesté-Lasserre[1], Nathalie Dupuy[1], Astrid Cannich[1], Luigi Bellocchio[1], Philippe Zizzari [1], Camille Allard [1], Delphine Gonzales [1], Yves Le Feuvre[1], Emeline Lhuillier [3], Alexandre Brochard [1], Jean Charles Nicolas[1], Jérémie Teillon[4], Macha Nikolski [5,6], Giovanni Marsicano [1], Xavier Fioramonti [7], Jens C. Brüning [2,8,9,10,11], Daniela Cota [1] & Carmelo Quarta [1] ✉

The hypothalamus contains a remarkable diversity of neurons that orchestrate behavioural and metabolic outputs in a highly plastic manner. Neuronal diversity is key to enabling hypothalamic functions and, according to the neuroscience dogma, it is predetermined during embryonic life. Here, by combining lineage tracing of hypothalamic pro-opiomelanocortin (Pomc) neurons with single-cell profiling approaches in adult male mice, we uncovered subpopulations of 'Ghost' neurons endowed with atypical molecular and functional identity. Compared to 'classical' Pomc neurons, Ghost neurons exhibit negligible *Pomc* expression and are 'invisible' to available neuroanatomical approaches and promoter-based reporter mice for studying Pomc biology. Ghost neuron numbers augment in diet-induced obese mice, independent of neurogenesis or cell death, but weight loss can reverse this shift. Our work challenges the notion of fixed, developmentally programmed neuronal identities in the mature hypothalamus and highlight the ability of specialised neurons to reversibly adapt their functional identity to adult-onset obesogenic stimuli.

The functional identity of mammalian brain neurons is determined during embryonic life by transcription factors and other molecules (referred as 'terminal selectors') that establish cell-specific genes and programs[1]. Neuron-specific functions are thought to be fixed once

developmental maturation is complete, and a neuron designed to promote food intake, for example, will always do so throughout life. However, this commonly accepted neurodevelopmental model is at odds with a poorly understood observation: terminal selectors are

[1]University of Bordeaux, INSERM, Neurocentre Magendie, U1215, F-33000 Bordeaux, France. [2]Department of Neuronal Control of Metabolism, Max Planck Institute for Metabolism Research, Cologne, Germany. [3]University of Toulouse III Paul Sabatier, INSERM, Institut des Maladies Métaboliques et Cardiovasculaires, U1297, 31400, France; GeT-Santé, Plateforme Génome et Transcriptome, GenoToul, Toulouse, France. [4]University of Bordeaux, CNRS, INSERM, BIC, US4, UAR 3420, F-33000 Bordeaux, France. [5]University of Bordeaux, Bordeaux Bioinformatics Center, Bordeaux, France. [6]University of Bordeaux, CNRS, IBGC UMR 5095 Bordeaux, France. [7]University of Bordeaux, INRAE, Bordeaux INP, NutriNeuro, UMR 1286, F-33000 Bordeaux, France. [8]Center for Endocrinology, Diabetes and Preventive Medicine (CEDP), University Hospital Cologne, Cologne, Germany. [9]Excellence Cluster on Cellular Stress Responses in Aging Associated Diseases (CECAD) University of Cologne, Cologne, Germany. [10]Center for Molecular Medicine Cologne (CMMC), University of Cologne, Cologne, Germany. [11]National Center for Diabetes Research (DZD), Neuherberg, Germany. ✉e-mail: carmelo.quarta@inserm.fr

active in some cases in post-mitotic mammalian neurons, such as principal forebrain neurons[1,2]. Thus, neurons that maintain expression of identity gatekeepers throughout life may use these signals to adapt cell identity and function to extracellular cues. Moreover, recent breakthroughs in single-cell mRNA sequencing (scRNAseq) and single-nucleus mRNA sequencing (snRNAseq) analyses of mature tissues are revolutionising the traditional understanding of how cell identity is delineated and governed across the lifespan of an organism. Within adult mammals, there is remarkable transcriptional heterogeneity among cells traditionally categorised as specific cell types that can exist in discrete and dynamically evolving states, a phenomenon that is intricately regulated by epigenetic pathways[3]. Whether (or not) hypothalamic neurons are endowed with such plasticity properties is currently unresolved, as our knowledge of the possible intrinsic factors involved is limited. Several intracellular pathways have been identified that control the fate of hypothalamic neurons during early embryonic life[4–8], but the constitutive signals that maintain and potentially reprogramme neuronal identity after development are more elusive, although recent advances have identified the T-box 3 gene (Tbx3)[9], Islet-1[10], Bsx[11], and microRNAs[12] as possible regulators. Within the arcuate nucleus (ARC) of the hypothalamus, a population of neurons expressing the neuropeptide precursor pro-opiomelanocortin (Pomc) is central to the regulation of food intake and energy balance[13–15] and it maintains functional levels of the identity factor Tbx3 throughout the entire lifespan[9]. Pomc neurons are highly heterogeneous at the molecular, functional, and spatial levels[16–19], reflecting the remarkable plasticity of *Pomc*-expressing cells during development. During embryonic differentiation, *Tbx3/Pomc* co-expressing precursors silence their identity marker *Pomc* and generate distinct neuronal types involved in neuroendocrine and metabolic control [agouti-related protein (AgRP)-or kisspeptin (Kiss)-expressing neurons][20,21], as well as Pomc neuron subtypes expressing different levels of *Pomc* and unique combinations of transcription factors[22]. This plasticity may persist into adulthood, as there is considerable cell-to-cell variability in the expression levels of the key identity marker *Pomc* within the mature hypothalamus[13,16]. Thus, adult *Pomc* expression levels may serve as a criterion to delineate subpopulations of Pomc neurons that are unique in their molecular and functional plasticity.

Here, we investigated this hypothesis by combining genetic lineage tracing of Pomc neurons in adult mice with multimodal single-cell profiling approaches. Using this approach, we uncovered subpopulations of 'Ghost' neurons with atypical molecular and functional identity and negligible *Pomc* expression. We also document that Ghost neurons are recruited in a mouse model of diet-induced obesity, independent of neurogenesis or changes in neuronal turnover. We conclude that prolonged exposure to dietary cues in adulthood alters mechanisms of neuronal identity maintenance, challenging the prevailing paradigm that developmentally programmed, cell-specific neuronal functions within the hypothalamus remain 'fixed' throughout adult life.

## Results

### Lineage tracing of Pomc neurons and identification of 'Ghost' neurons

To track Pomc neurons throughout adult life, independent of their actual *Pomc* expression, we crossed mice expressing the tamoxifen (TAM)-inducible CreERT2 recombinase under specific *Pomc* promoter elements[23] with two different lineage tracing reporters [ROSA-tdTomato (Pomc$^{CreERT2}$;tdTomato) or ROSA-ZsGreen (Pomc$^{CreERT2}$;ZsGreen)]. Using fluorescent in situ hybridisation (FISH) and immunohistochemistry (IHC) followed by single cell image analysis, we detected 'classical' Pomc neurons that were positive for both the reporter protein and *Pomc* (hereafter referred to as Pomc + ) in the ARC of adult chow diet (CD)-fed reporter male mice (Fig. 1A, B). Notably, we also identified a subset of cells that were negative for both *Pomc* mRNA and Pomc protein in adulthood, but expressed the Pomc lineage tracer

(Fig. 1A, B). Comparable relative distributions of Pomc+ vs Pomc-traced cells were found in male and female mice (Fig. S1A). Single molecule mRNA FISH (smFISH, RNA Scope™) confirmed the absence of *Pomc* mRNA signal in ~10% of these tracked neurons (Fig. S1B). Previous studies have identified mature Pomc cells with low *Pomc* mRNA expression[16,24] but there has been no evidence of Pomc neurons that are completely invisible for their main functional marker during adult life, leading us to refer to these neurons as 'Ghost' cells.

To exclude that Ghost cells are the result of confounding off-target genetic effects, we performed a series of validation experiments. First, we analyzed the co-expression of ZsGreen and *Pomc* mRNA in Pomc$^{CreERT2}$;ZsGreen mice treated with TAM versus vehicle. No ZsGreen signal was detected in vehicle-treated controls, whereas most ZsGreen cells in TAM-treated animals were co-positive for *Pomc* (Fig. S1C), indicating absence of basal CRE activity. The average number of ARC Pomc-expressing neurons did not change after TAM treatment (Fig. S1C), ruling out confounding TAM-mediated off-target effects on *Pomc* expression. We also assessed CRE expression in the whole brain of TAM-treated Pomc$^{CreERT2}$;ZsGreen mice using the tissue clearing technique CUBIC[25] in combination with light sheet fluorescence microscopy (LSFM). The reporter signal was exclusively localised within the mediobasal hypothalamus, consistent with the known anatomical profile of Pomc neurons in the forebrain (Supplementary Video 1).

After crossing Pomc$^{CreERT2}$;tdTomato mice with animals expressing the green fluorescent protein (eGFP) under the *Pomc* promoter[14] (Pomc$^{CreERT2}$;tdTomato;Pomc-eGFP), we successfully detected tdTomato-positive/eGFP-negative (Ghost) cells in this alternative model (Fig. 1C). Ghost cells were also detected using a lineage model in which the recombinase DRE is specifically expressed postnatally in a subset of bona fide Pomc neurons[17] (Fig. S1D), further confirming that they are not an off-target artefact.

Traces of *Pomc* mRNA expression, however, were notably found in Ghost cells isolated from Pomc$^{CreERT2}$;tdTomato;Pomc-eGFP mice using a highly sensitive approach combining specific *Pomc* mRNA pre-amplification with single cell qPCR (Table S1), as confirmed by DNA sequencing of the amplification product (Supplementary file 1). We conclude that the negligible levels of *Pomc* mRNA in Ghost cells prevent their detection by conventional neuroanatomical approaches and commonly used *Pomc* promoter-based transgenic lines. However, this very low level of *Pomc* expression is sufficient to specifically activate the lineage reporter in these cells.

We then asked whether the number of Ghost cells might be dynamically regulated throughout life under physiological conditions. In reporter Pomc$^{CreERT2}$;tdTomato animals treated with 1 versus 2 consecutive cycles of TAM injection, we observed comparable numbers of Ghost cells (Fig. S1E). Time titrations at day 1, 4 and 14 after TAM injection in the same model showed no difference in the relative number of Ghost cells (Fig. S1F). The same number of Ghost cells was found in 5-week-old mice exposed to lineage tracing (Fig. S1G) compared to adult (12-week-old) animals (Fig. S1E), indicating that these cells are born early in life and remain stable under physiological conditions. These results also show that, although systemic levels of TAM may persist for a prolonged period following after treatment interruption, as has recently been suggested[26], in our model the recombination of Pomc neurons induced by TAM administration peaks a few days after treatment and no additional TAM-mediated effects are observed at subsequent time points.

### Molecular, spatial and functional profiling of 'Ghost' neurons

To elucidate the molecular nature of Ghost cells, we examined the co-expression of markers that define the functional identity of Pomc neurons. A reduced number of Ghost cells co-expressed the identity markers cocaine-and amphetamine-regulated transcript (*Cart*) compared to Pomc+ neurons (Fig. 1D and S2A). The identity gatekeeper

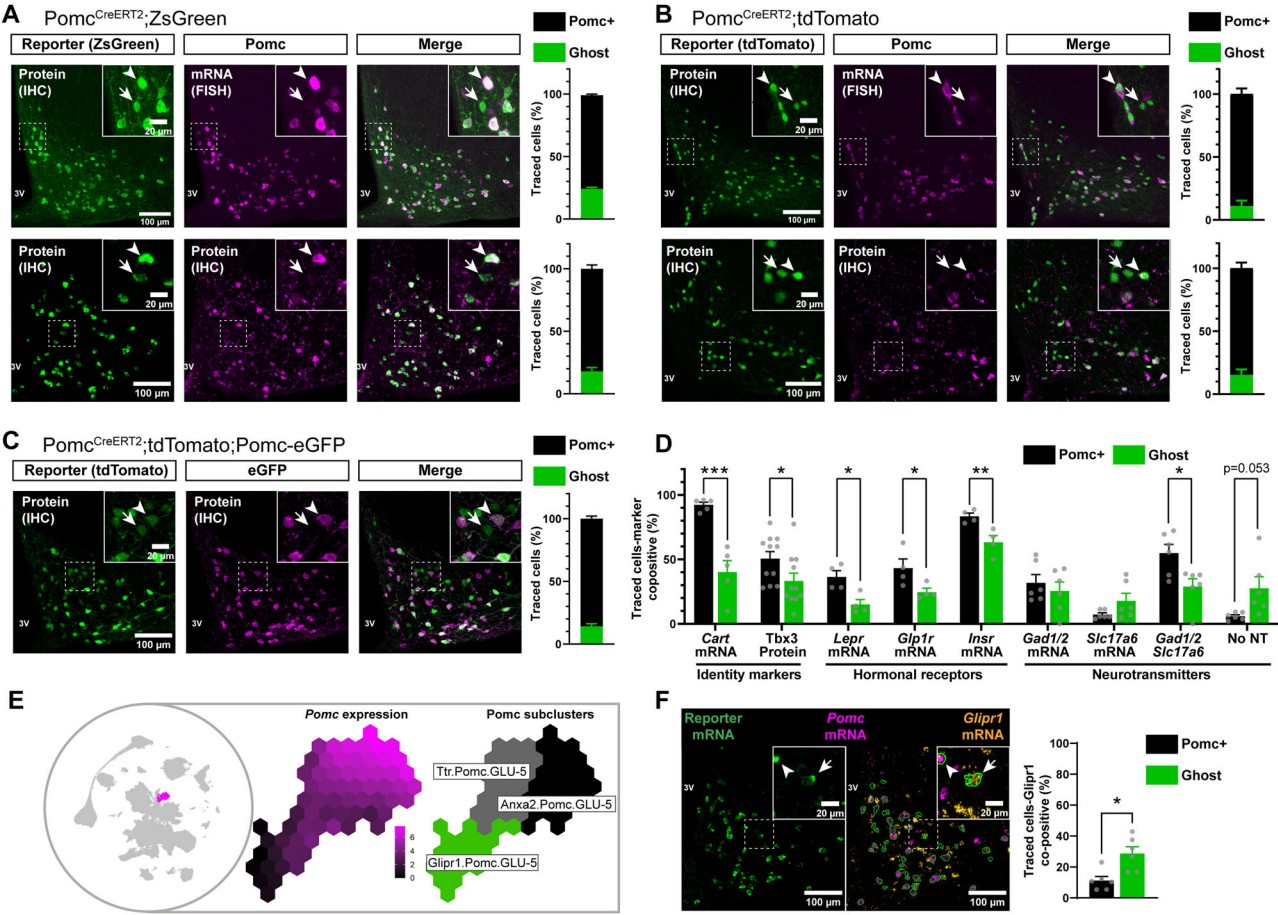

**Fig. 1 | Lineage tracing of Pomc neurons in adult chow-fed mice reveals subtypes with atypical molecular identity. A** Detection and quantification of Pomc+ and Ghost neurons in Pomc[CreERT2];ZsGreen male mice by analyzing reporter cells (ZsGreen) positive (Pomc + ) or negative (Ghost) for *Pomc* mRNA (FISH, $n = 11$ mice) or Pomc protein (IHC, $n = 5$ mice). **B** Detection and quantification of Pomc+ and Ghost neurons in Pomc[CreERT2];tdTomato male mice by analyzing reporter cells (tdTomato) positive (Pomc + ) or negative (Ghost) for *Pomc* mRNA (FISH, $n = 3$ mice) or Pomc protein (IHC, $n = 4$ mice). **C** Detection and quantification of Pomc+ and Ghost neurons in Pomc[CreERT2];tdTomato;Pomc-eGFP male mice ($n = 4$) by analyzing reporter cells (tdTomato) positive (Pomc + ) or negative (Ghost) for endogenous eGFP protein by IHC. **D** Quantification of Pomc+ and Ghost neurons co-expressing detectable mRNA or protein levels of several Pomc neuronal functional markers in Pomc[CreERT2];ZsGreen or Pomc[CreERT2];tdTomato male mice. *Cart* mRNA was assessed by smFISH ($n = 5$ mice), Tbx3 by IHC ($n = 11$ mice), hormone receptors by smFISH ($n = 4$ mice), and neurotransmitter-related markers by smFISH ($n = 6$ mice). **E** Pomc neurons in the hypothalamus by the scRNAseq atlas HypoMap. Left: Global Uniform Manifold Approximation and Projection (UMAP) with Pomc-cluster position highlighted.

Middle: Summarised *Pomc* expression levels within the POMC-cluster. Right: Pomc neuron subclusters. **F** Representative image and quantification of the percentage of Pomc+ and Ghost neurons co-postive for *Glipr1* mRNA by smFISH in Pomc[CreERT2];tdTomato male mice ($n = 6$). *Pomc:* pro-opiomelanocortin, *AgRP:* agouti-related protein, *Npy:* neuropeptide y, *Kiss:* kisspeptin, *Cart:* cocaine-and amphetamine-regulated transcript, *Tbx3:* T-box gene 3, *Lepr:* leptin receptor, *Glp1r:* glucagon-like peptide-1 receptors *Gad1/2:* Glutamate decarboxylase 1 and 2, S*lc17a6*: vesicular glutamate transporter 2, IHC: immunohistochemistry, FISH fluorescence in situ hybridisation, smFISH single molecule fluorescence in situ hybridisation, NT neurotransmitter. *Glipr1*: Glioma pathogenesis-related protein 1. Pomc+ neurons are indicated by arrowheads, Ghost neurons by arrows. Data in (**A−D**, and **F**) are mean ± s.e.m. from 3 independent experiments. n indicates the individual biological values. Data in (**A−D** and **F**) were obtained 2 weeks after adult onset (12-week-old mice) administration of tamoxifen. In **D**: ***$P = 0.0004$ (*Cart*), *$P = 0.0453$ (Tbx3), *$P = 0.0129$ (*Lepr*), *$P = 0.049$ (*Glp1r*), **$P = 0.0117$ (*Insr*),*$P = 0.016$ (*Slc17a6*) by two-tailed $t$ test. In **F**, *$P = 0.0154$ by two-tailed t-test. Source data are provided as a Source Data file.

*Tbx3* was also expressed in a lower fraction of Ghost cells (Fig. 1D and S2A). Similarly, leptin receptors (*Lepr*), insulin receptors (*Insr*) and glucagon-like peptide-1 receptors (*Glp1r*), which underlie POMC neuronal functions in response to hormones[27], were all expressed in lower proportion in Ghost neurons (Fig. 1D, S2A). Different Pomc neuron subsets may modulate neurotransmission by secreting either the inhibitory neurotransmitter GABA, the excitatory neurotransmitter glutamate, or both[16,28,29]. Ghost cells did not present relative changes in the co-expression of the GABAergic marker *GAD65/67* (*Gad1/2*) or the glutamatergic marker *VGlut2* (*Slc17a6*) compared to Pomc+ neurons. However, a lower proportion of atypical neurons were positive, and a higher number tended to be negative for these markers when expressed together (Fig. 1D, S2A). Previously described populations of Pomc cells expressing low levels of *Pomc* mRNA are enriched for *AgRP* and neuropeptide y (*Npy*) mRNAs[16], but we did not observe

differential *AgRP* and *Npy* expression in Ghost neurons (Fig. S2B). As Pomc progenitors can generate Kiss neurons between alternative lineages, we further examined *Kiss* mRNA co-expression in Ghost and Typical cells, but found no evidence for co-expression of this marker (Fig. S2C).

Our data reveal that Ghost cells have an atypical molecular identity and may represent molecularly distinct neuronal subsets that have remained so far 'hidden' in the field. To further explore this possibility, we compared molecular markers differentially expressed in Ghost cells (Fig. 1D) with a comprehensive scRNAseq and snRNAseq gene expression atlas of the mouse hypothalamus (HypoMap[30]) to potentially identify whether Ghost-like molecular signatures overlap with previously described neuronal subtypes. The Pomc-expressing cluster (Pomc.GLU-5) could be divided into three distinct subpopulations (Fig. 1E, S2D-F). *Pomc* and other genes defining the Ghost cell profile

(*Glp1r*, *Lepr*, *Cartpt* and *Tbx3*, Fig. 1D) were expressed at lower levels in one of these clusters defined by GLI pathogenesis related 1 (*Glipr1*) expression (Fig. 1E and S2F), leading us to predict that cells from this cluster might have Ghost-like properties. smFISH-based image analysis of *Glipr1* co-expression in Pomc+ and Ghost cells in reporter Pomc[CreERT2];tdTomato mice revealed a higher proportion of *Glipr1*-positive Ghost cells compared to control Pomc+ cells (Fig. 1F). However, a noteworthy proportion of Ghost cells (70%) did not exhibit co-expression with this marker. Thus, while a specific fraction of Ghost cells are aligned with the described Glipr1 cluster, a substantial number of these neurons may represent additional, hitherto unidentified molecular clusters. Accordingly, existing scRNAseq atlases of the adult mouse hypothalamus[13,16,24,30,31] have not been generated in combination with lineage tracing of adult Pomc neurons, and, in the absence of lineage reporter markers, cells with too low *Pomc* mRNA levels may have 'escaped' these published databases due to sensitivity limitations and the so-called 'dropout effect'[32].

Different Pomc neuronal clusters exhibit highly heterogeneous spatial positions[13,17]. To obtain a holistic 3D representation of Ghost vs Pomc+ neurons, we used CUBIC[25] in combination with LSFM in CD-fed reporter Pomc[CreERT2];tdTomato;Pomc-eGFP mice, and analyzed the spatial data as previously described[17] (Fig. 2A–D and S3). We observed significant differences in the spatial localisation pattern of Pomc+ neurons compared to Ghost cells (Fig. 2E), with the latter more predominantly located in the caudal ARC region (Fig. 2D, E), as confirmed by cumulative cell distribution analysis (Fig. 2F), supporting our hypothesis that Ghost cells have a distinct spatial distribution.

Pomc neurons regulate energy balance in response to the energy needs of the body and changes in peripheral hormones[27]. Re-feeding after prolonged fasting increases Pomc neuronal activity, as previously assessed by the activation marker c-Fos and calcium imaging[9,19,33]. To investigate the functional properties of Ghost cells, we subjected CD-fed Pomc[CreERT2];tdTomato mice to fasting (24 h) or refeeding (2 h) with a high fat diet (HFD) and assessed neuronal c-Fos expression. Re-feeding resulted in a significantly increased number of Pomc+ neurons co-expressing c-Fos compared to fasting (Fig. 3A, B), consistent with previous observations[33,34], but there was no evidence of Ghost cell activation (Fig. 3A, B), regardless of changes in the relative number of Ghost vs. Pomc+ neurons (Fig. S4A).

We then examined the response of Pomc neurons to the metabolic hormone leptin in reporter mice 45 min after the administration of the hormone. A significant number of Pomc+ neurons showed increased Lepr signalling as assessed by Stat3 phosphorylation (pStat3), but a significantly smaller number of Ghost cells were positive for this marker (Fig. 3C, D).

To explore dynamic activity properties, we took advantage of the Pomc[CreERT2];tdTomato;Pomc-eGFP model, which allows the detection of Ghost cells based on endogenous fluorescence (Fig. 1C), and followed longitudinal changes in electrophysiological responses to leptin and insulin using ex vivo patch clamp recordings. No differences in intrinsic properties or action potential properties were observed when comparing Pomc+ and Ghost neurons before hormone challenge (Fig. S4B), indicating that Ghost cells have mature neuron-like features. Single neuron analysis, performed by comparing the resting membrane potential and input resistance before and during leptin application, revealed three types of Pomc+ neurons: (1) those activated by leptin (33%); (2) inhibited cells (38%); and (3) unresponsive cells (29%) (Fig. 3E), confirming our previous observations[24]. However, most Ghost cells were unresponsive (71%, Fig. 3E). Compared to Pomc+ neurons, Ghost cells also showed a different response pattern to insulin, with a higher number of cells unresponsive to the hormone and a larger fraction showing inhibition (Fig. 3F). Thus, Ghost neurons are less sensitive to nutritional and hormonal cues involved in the regulation of energy balance.

## Diet-induced obesity leads to recruitment of 'Ghost' neurons

Diet-induced obesity (DIO) is associated with dysregulated neuronal responses to metabolic hormones[27] and maladaptive changes in the identity maintenance of peripheral organs such as endocrine pancreas and adipose tissue[35,36]. To investigate whether changes in cell identity may also affect Pomc neurons during DIO, we quantified the number of Ghost cells in our reporter mice exposed to an obesogenic HFD for 3 or 6 months (Fig. 4A). After 6 months of HFD, DIO mice had an increased number of Ghost cells, as demonstrated by either FISH (*Pomc* mRNA, Fig. 4B), smFISH (*Pomc* mRNA, Fig. 4C) or IHC (Pomc protein, Fig. 4D). Representative images for these analyses are shown in Fig. S5A−C. The observed changes in Ghost cell numbers in DIO mice fed with HFD for 6 months were independent of variations in total reporter-positive cells (Fig. S5D). Compared to CD controls, no difference in the number of Ghost versus Pomc+ neurons were observed after 3 months of HFD in Pomc[CreERT2];ZsGreen mice (Fig. S5E, F).

A positive correlation was found between Ghost cell number and body weight in DIO mice (Fig. S5G), suggesting that Ghost cell recruitment in obesity is associated with differential sensitivities to diet-induced weight gain. Remarkably, the recruitment of atypical neurons could be rescued by switching these DIO mice back to CD for 4 weeks (Fig. 4D), resulting in significant weight loss (Fig. S5H). Thus, these data demonstrate that changes in caloric intake and body weight induce reversible and plastic changes in the number of Ghost neurons.

No effects of DIO on the cell proliferation marker KI67 were observed, as it was virtually unexpressed in tracked neurons under both CD and HFD conditions (Fig. S6A). Although a few tracked cells were positive for the apoptotic marker activated Caspase-3, there was no differential co-expression between CD and HFD mice (Fig. S6B).

Consistent with previous findings[37,38], short-term HFD feeding (3 weeks) resulted in an increase in the number of proliferating cells in the ARC, as assessed by intracerebroventricular (ICV) infusion of BrdU in our reporter mice (Fig. S6C). A small fraction (5%) of these proliferating cells were Pomc+ neurons, as previously shown[39,40], but no BrdU-positive Ghost cells were found in either HFD or CD mice (Fig. S6C). To determine the effects of long-term HFD feeding on cell turnover, we subjected a second cohort of mice to continuous ICV BrdU infusion from month 3 of HFD until month 6, when greater numbers of Ghost cells was observed. Exposure to HFD for 6 months increased the total number of Pomc-tracked neurons co-expressing BrdU compared to the previous short-term HFD paradigm (Fig. S6D). However, we did not observe differential BrdU accumulation in the Ghost vs. Pomc+ populations after such a long HFD feeding period (Fig. S6D). In this new group of BrdU-treated mice, the total number of Pomc cells tracked did not change after 6 months of HFD exposure compared to the CD condition (Fig. S6D), confirming our previous observation (Fig. S5D). Considering also that we found no evidence of HFD-induced cell death (Fig. S6B), a possible interpretation of these collective findings is that Pomc precursors tracked during TAM administration can generate both new Ghost and new typical Pomc neurons to the same extent after long-term HFD. However, some typical tracked Pomc neurons loose their initial molecular identity over time and become Ghost neurons in response to the overnutrition cue.

Our data, therefore, reveal that the increased number of tracked Ghost cells in DIO mice is not due to previously described mechanisms linking obesity with neuronal dysfunction, such as neuronal death[41,42] or altered neurogenesis[43,44] and point to a mechanism of molecular and functional conversion of typical Pomc-to-Ghost neurons in obesity. To investigate potential mechanisms underlying such phenotypic conversion, we examined several markers of differentiation and maturation. An overall lower number of Ghost cells were positive for the maturation marker NeuN compared to Pomc+ neurons, independently of the diet (Fig. S6E). Two additional markers of immature (Sox2) or mature (Calbindin-D28K) neurons showed no differential expression in Ghost vs typical Pomc+ neurons, regardless of diet (Fig. S6F, G). Long-

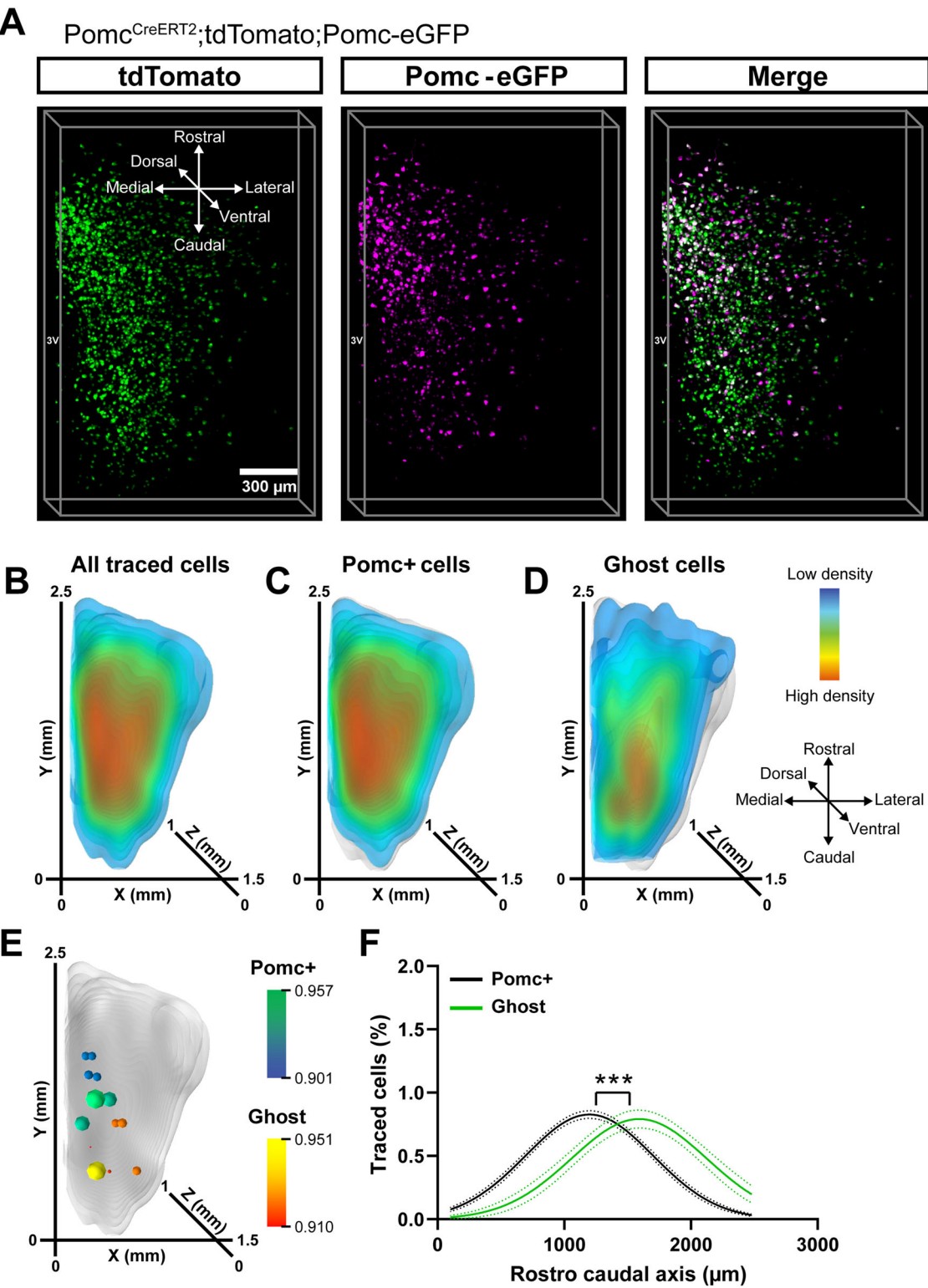

**Fig. 2 | Ghost neurons show a distinct spatial distribution under chow-diet.**
**A** Representative 3D reconstruction of populations expressing tdTomato, Pomc-eGFP (hemi-arcuate nucleus), and the combined signal in Pomc^CreERT2;tdTomato;Pomc-eGFP male mice ($n = 3$) 2 weeks after adult onset (12-week-old mice) administration of tamoxifen. Isosurface density plots of the whole traced cell population (**B**), the Pomc+ subset (**C**) and Ghost neurons (**D**). Grey shaded areas in (**D**, **E**) represent the whole traced cell population. **E** Statistical representation of the differences in distribution between the Pomc+ cells and Ghost cells using a two-tailed *t* test. *P* values are plotted as spheres within the space occupied by the neurons (background). The size and colour of the spheres indicate the significance values in a range from green to blue (Pomc + ) and yellow to red (Ghost). **F** Cumulative distribution of the Pomc+ and Ghost subpopulations for the rostro-caudal spatial axis. Data are shown as non-linear fitted curves (Gaussian). ***$P < 0.001$ with extra-sum-of-squares F-test comparison. Data were obtained from 2 independent experiments. n indicates the individual biological values. Source data are provided as a Source Data file.

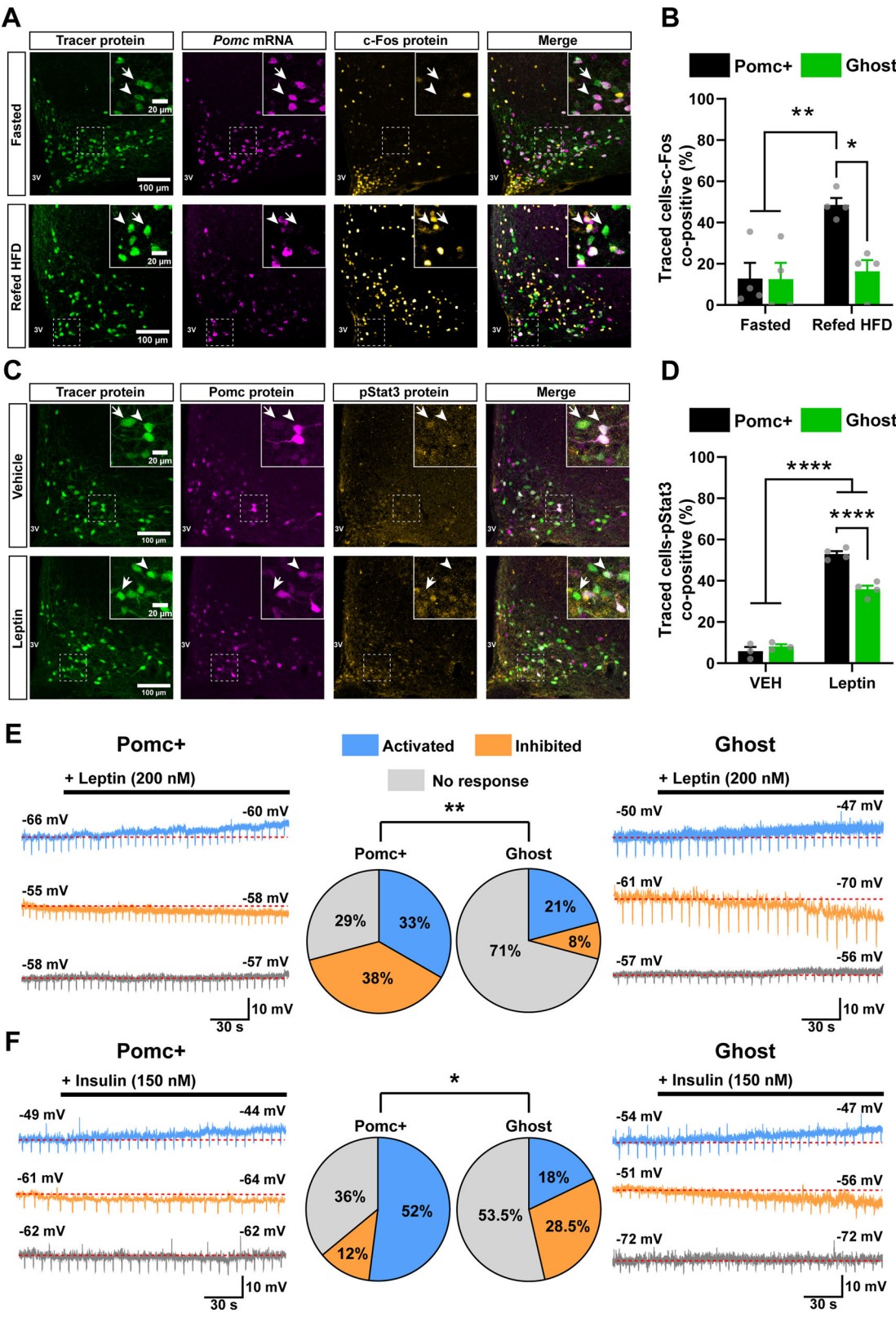

term HFD consumption reduced the expression of calbindin in both cell groups (Fig. S6F). These findings rule out the possibility that Ghost cells have progenitor-like properties and may represent previously described *Pomc*-expressing precursors endowed with identity plasticity[20,21], which is consistent with our electrophysiological data showing that Ghost cells have mature neuron-like properties (Fig. S4B) and with the lack of co-expression of the alternative fate markers AgRP,

Npy and Kiss (Fig. S2B, C). These data also imply that the recruitment of Ghost cells in obesity may not involve neuronal dedifferentiation.

## Unbiased demonstration of 'Ghost' neurons recruitment in obesity

To provide an unbiased support for our model of Pomc typical-to-Ghost neuron conversion, we combined ex vivo patch clamp

**Fig. 3 | Ghost cells have atypical sensitivity to nutritional and hormonal cues.**
**A** Representative FISH/IHC images of the tdTomato reporter (protein), *Pomc* (mRNA) and c-FOS (protein) in Pomc^CreERT2;tdTomato male mice challenged with fasting (24 h, *n* = 4) or fasting followed by refeeding (2 h, *n* = 4). **B** Quantification of Pomc+ versus Ghost neurons positive for c-FOS relative to (**A**). **C** Representative IHC images of the tdTomato reporter, Pomc protein, and pStat3 in CD-fed Pomc^CreERT2;tdTomato male mice injected subcutaneously with vehicle (*n* = 3) or leptin (5 mg/kg; *n* = 4) 45 min before brain harvest. **D** Quantification of the percentage of reporter cells (tdTomato) positive for pStat3 in Pomc+ versus Ghost neurons relative to (**C**). **E, F** Ex-vivo whole cell patch-clamp recording of Ghost and Pomc+ neurons in Pomc^CreERT2;tdTomato;Pomc-eGFP male mice following leptin (200 nM) or insulin (150 nM) challenge. Representative traces of Pomc+ or Ghost neurons are shown on the left and right. Middle: % of activated cells (Δ resting membrane potential ≥+2 mV, Δ input resistance ≥10% change) in response to leptin (**E**, blue; Pomc + : *n* = 8; Ghost: *n* = 5) or insulin (**F**, blue; Pomc + : *n* = 13; Ghost: *n* = 5),

inhibited cells (Δ resting membrane potential ≤−2 mV, Δ input resistance ≥10% change) in response to leptin (**E**, orange; Pomc + : *n* = 9; Ghost: *n* = 2) or insulin (**F**, orange; Pomc + : *n* = 3; Ghost: *n* = 8) and non-responsive cells (grey; Leptin: Pomc + : *n* = 7; Ghost: *n* = 17; insulin: Pomc + : *n* = 9; Ghost: *n* = 15). Downward deflections in current-clamp recordings represent the membrane voltage responses to constant hyperpolarizing currents. Dotted lines represent the resting membrane potential. 16 total mice for leptin and 17 for insulin were analyzed in this experiment. pStat3: phosphorylated transducer and activator of transcription-3, CD: chow diet, HFD: high fat diet. In (**A**, **C**) arrowheads define Pomc+ neurons, arrows define Ghost neurons. All mice were studied 2 weeks after adult onset (12-week-old mice) administration of tamoxifen. n indicates the individual biological values. Data are mean ± s.e.m from 2 independent experiments and were analyzed by 2-way ANOVA followed by Tukey's post-test. In (**B**), **P = 0.0176 and *P = 0.091. In (**D**), ****P < 0,0001. Data in (**E**, **F**) were obtained from 3 independent experiments analyzed by chi-squared test. Source data are provided as a Source Data file.

recordings with scRNAseq of the patched cell (Patch-seq) from reporter Pomc^CreERT2;tdTomato;Pomc-eGFP mice exposed to CD or 6 months HFD (patch-seq). We pre-selected *n* = 34 ghost cells (15 chow vs 19 HFD) and *n* = 42 Pomc+ (23 chow vs 19 HFD) from these animals based on the endogenous presence or absence of the fluorescent Pomc-eGFP signal. Using unbiased cluster analysis of the mRNA profile of all cells, we observed that pre-selected Ghost cells formed distinct molecular clusters (a and d) compared to typical Pomc+ cells (b and c, Figs. 4E, S7A). The molecular profile of the different clusters was not affected by differential expression of ribosomal or mitochondrial genes (Fig. S7B), nor by differences in the number of reads (Fig. S7C), supporting the quality of the analysis.

While the Pomc+ clusters 'b' and 'c' exhibited the classic molecular phenotype of Pomc neurons, with high levels of *Pomc* and *Tbx3* mRNA, and high expression of receptors for metabolic hormones (Fig. 4F), the Ghost clusters 'a' and 'd' had similar molecular properties (Fig. 4F), compared to our previous neuroanatomical analyses (Fig. 1). Likely due to low basal mRNA expression levels and the sequencing depth used, the previously identified target *Glipr1* was not detectable in our patch-seq dataset.

After matching our mRNA-seq data with specific Pomc neuron markers from HypoMap[30], it was striking that both Ghost neuron clusters identified by Patch-seq had reduced expression of most previously known Pomc-specific genes (Fig. 4G), providing unbiased confirmation of the existence of Ghost Pomc neurons with atypical molecular identity. The number of cells belonging to the Ghost-like cluster 'd' was increased in HFD-fed mice compared to CD controls, whereas cells from the typical cluster 'c', were reduced (Fig. 4H), suggesting recruitment of a fraction of Ghost neurons in obesity, in parallel with reduced numbers of typical neurons. Significant changes in electrophysiological parameters of the Ghost cluster 'a' were particularly noted in HFD animals, consistent with increased neuronal excitability, but not in the other identified clusters (Fig. 4I), demonstrating that prolonged dietary cues induce selective activity changes in Ghost-like neurons.

## Discussion

In this study, we uncover and characterise subpopulations of 'Ghost' neurons with atypical identity within the hypothalamus, shedding light on their distinctive molecular and functional profiles. Our results indicate that these neurons are recruited in obesity and exhibit specific changes in activity under this condition. Rather than simply being Pomc neurons that modulate *Pomc* expression, Ghost neurons are likely to represent Pomc subtypes with unique molecular, spatial and functional properties that have not been previously identified, challenging existing paradigms of hypothalamic neuronal diversity and its implications for metabolic regulation.

The recruitment of Ghost neurons in our animal model of obesity appears to be independent of changes in the total number of reporter-

positive cells, proliferation markers, apoptosis markers or dedifferentiation markers. Thus, this phenomenon is likely to result from (mal)adaptive mechanisms involving the phenotypic transformation of specialised Pomc subsets, sensitive to dietary and hormonal fluctuations, into Ghost-like neurons. However, our findings are not accompanied by changes in the canonical cell fate determination mechanisms known to govern immature cells. Therefore, it remains to be definitively established whether Ghost neurons in obesity can be considered a hallmark of a postnatal 'identity switch'. Nonetheless, recent advances in the analysis of single cells within mature tissues are challenging the conventional understanding of how cell identity is regulated throughout the life of an organism[3]: mature cells in different tissues, despite sharing similar morphological characteristics, are likely to exhibit diverse and dynamically changing molecular profiles, even independently of their differentiation status[3]. In the context of this evolving paradigm, specific adult Pomc neurons may have the ability to dynamically lose and regain their developmentally programmed functions in response to prolonged changes in body weight, independent of changes in differentiation programs. However, the intrinsic heterogeneity observed in Ghost cells and the relatively modest sample size (76 cells) in our patch-seq analyses hinder a comprehensive analysis of this model through dynamic evaluation of cell fate trajectories. Therefore, further investigations using larger single cell datasets alongside Pomc neuronal lineage tracing will be essential to elucidate the potential of specialised Pomc neurons to undergo molecular identity changes in response to obesity.

Our data show that certain molecularly defined sub-clusters of atypical Ghost neurons increase in number in obesity, while others are not recruited but show selective changes in activity in response to obesogenic cues. This suggests that distinct subpopulations of Pomc Ghost neurons may undergo differential changes in properties under obesogenic conditions and illustrates that the remarkable intrinsic heterogeneity of hypothalamic Pomc neurons is further enhanced under obesogenic conditions.

Obesity is a neuroendocrine disease characterised by dysregulated peripheral levels of metabolic hormones and impaired counter-regulatory neuronal responses to these signals[45]. The recruitment of 'invisible' Ghost neurons observed in DIO mice may reflect an adaptive mechanism whereby the brain attempts to cope with a continuous energy surplus, but ultimately fails to trigger feedback signals that counteract disease progression and hormonal dysfunction. Thus, the accumulation of these atypical Pomc-derived cells may explain the described impaired ability of hypothalamic Pomc neurons to respond to peripheral metabolic hormones during diet-induced metabolic stress[27,45].

Exposure to obesogenic diets in adulthood, both in preclinical models and in humans suffering from obesity, has been associated with altered cell identity maintenance of peripheral tissues such as adipocytes[36] or pancreatic β-cells[35], primarily through the mechanism

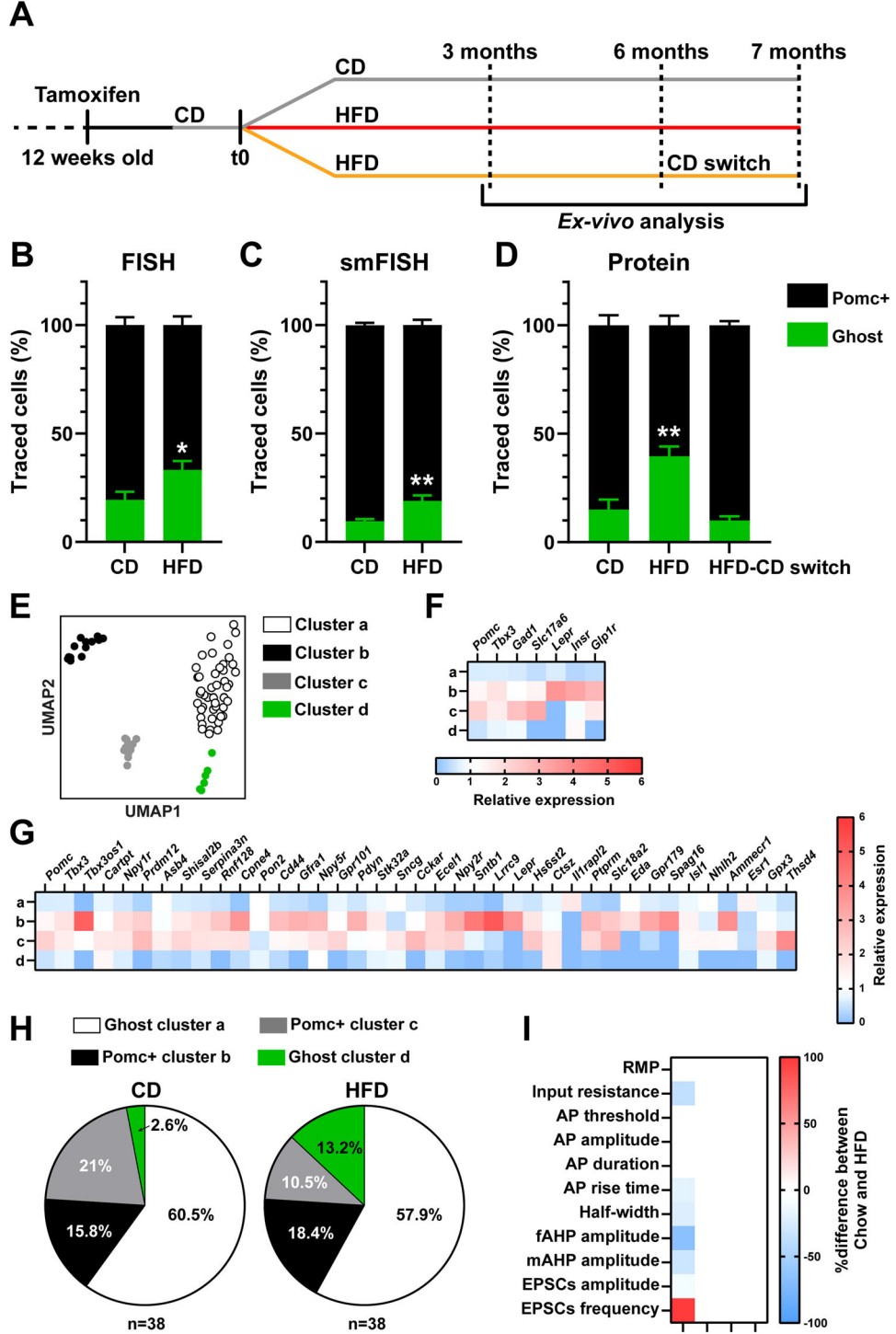

of de/transdifferentiation. Nevertheless, the phenotypic switch observed in the current study may potentially follow different molecular pathways and cellular mechanisms than those previously reported for peripheral non-neuronal cells or embryonic progenitors, highlighting the nuanced and context-specific nature of cellular responses to obesogenic stimuli, particularly within the neural tissue. The essential intrinsic molecular hubs that control the maturational state of adult hypothalamic neurons remain elusive. Given the paucity of markers of postnatal maturation/differentiation in the hypothalamus and the remarkable cell diversity within this particular brain region[13,16,24,30,31], it is premature to dismiss the possibility that specific Pomc neuron subtypes may alter cellular phenotypes via de/

transdifferentiation mechanisms. The recruitment of Ghost neurons in obesity may also involve epigenetic mechanisms that are essential for both the maintenance of neuronal fate[1,2] and the regulation of energy balance[46–49]. Furthermore, alterations in the activity of the identity gatekeeper Tbx3, as evidenced by the reduced expression of this marker in Ghost cells compared to typical Pomc neurons, may potentially contribute to our observations.

The observed variability in the detection rates of Ghost neurons among the different models employed is a limitation of our study. Nonetheless, the consistent detection of these neurons with atypical identity across various independent models and experimental paradigms suggests that the observed biological phenomenon is not

**Fig. 4 | Ghost cell recruitment in HFD-induced obesity. A** Reporter mice were treated with tamoxifen at 12 weeks of age and exposed to HFD, CD or (HFD-CD switch) for different durations 2 weeks later (t0). **B, C** Quantification of Pomc+ and Ghost neurons in Pomc^CreERT2;ZsGreen male mice fed with CD or HFD (6 months). *Pomc* mRNA was analyzed by FISH (**B**, *n* = 8 per group) or smFISH (**C**, *n* = 6 per group). **D** Quantification by IHC of Pomc+ and Ghost neurons in Pomc^CreERT2;ZsGreen male mice fed with CD (*n* = 4), HFD (7 months, *n* = 4), or exposed to HFD-CD switch (*n* = 3). **E** UMAP representation of 76 neurons analyzed by patch-seq in Pomc^CreERT2;tdTomato male mice exposed to CD (*n* = 5) or HFD (*n* = 5) for 6 months. We collected *n* = 38 cells from CD mice and *n* = 38 from HFD mice. We identified 4 main clusters: a (45 cells), b (13 cells), c (12 cells) and d (6 cells). **F** Heatmap of the relative gene expression (mean expression one cluster relative to the others) of Typical Pomc+ markers altered in Ghost neurons based on the data in Fig. 1D. **G** Heatmap of the relative gene expression (mean

expression of one cluster relative to the others) of Pomc-neuron specific genes from HypoMAP in the clusters identified in (**E**). **H** Change in the % of cells in each cluster between the CD and HFD conditions. **I** Effects of HFD exposure (6 months) on electrophysiological parameters in the different clusters that were significantly different (HFD vs. CD) by two-tailed *t* test. RMP resting membrane potential. AP action potential. fAHP fast afterhyperpolarization. mAHP medium afterhyperpolarization. EPSCs: excitatory post synaptic currents. HFD high fat diet, CD chow diet. Data in (**B–D**) are mean ± s.e.m from 3 (**B, C**) or 2 (**D**) independent experiments. Data in (**E**) were obtained from 3 independent experiments. n indicates the individual biological values. In (**B, C**) *$P$ = 0.0207 and **$P$ = 0.0087 by two-tailed *t* test (CD Ghost vs HFD Ghost). In (**D**), **$P$ = 0.0058 (CD Ghost vs HFD Ghost) and **$P$ = 0.0022 (HFD Ghost vs HFD-CD Ghost) by ANOVA followed by Tukey's post-test. Data in (**H**) are shown as a percentage of the total population $P$ = 0,0004 by chi-squared test. Source data are in the Source Data file.

stochastic. A reduced number of Ghost neurons was found when using the Pomc-Dre line compared to the Pomc^CreERT2 model (5% vs 20%), which can be explained by the lower total number of Pomc neurons showing Dre-mediated recombination[17] (Fig. S1D). The inconsistency in the number of Ghost cells detected between ROSA-ZsGreen vs. ROSA-tdTomato models (20% vs. 10% of Ghost neurons) may instead be related to the different efficiency of the IHC-based analysis of the different protein reporters. About 20% of neurons are positive for Pomc but negative for the lineage marker (see Fig. 1). Therefore, the efficiency of the induced recombination is sub-maximal in animal models commonly used to study Pomc neuronal biology. However, this caveat does not undermine the validity of our observations, as our focus was on cells positive for the lineage tracer, although the absolute number of Ghost neurons may have been underestimated.

In conclusion, our findings challenge the dogma that Pomc neuronal functional identities are 'fixed' after development and provide a conceptual framework for understanding whether and how mechanisms of neuronal identity plasticity in the brain may contribute to energy balance dysregulation and the development of obesity.

## Methods

### Ethical approval
All experiments were conducted in strict compliance with the European Union recommendations (2013/63/EU) and were approved by the French Ministry of Higher Education, Research and Innovation and the local ethical committee of the University of Bordeaux (APAFIS 2018011815151678, APAFIS 2019092009514330, APAFIS 2019102509231937).

### Mice
Unless otherwise stated, all experiments were performed on male mice. Mice were individually housed in standard plastic rodent cages, maintained at 22 ± 2 °C on a 12-h light-dark cycle (lights off at 13:00 h, 45–65% humidity) with ad libitum access to pelleted chow diet (CD, Standard Rodent Diet A03, SAFE, France; 3.236 kcal/g; 13.5% calories from fat, 25.2% calories from protein and 61.3% calories from carbohydrate) or a high-fat diet (HFD, D12492; Research Diets, New Brunswick, NJ; 60% calories from fat, 20% calories from protein and 20% calories from carbohydrate). In the experiments where we tested the effects of diet-induced obesity, mice were fed either CD or HFD for 3 or 6 months prior to brain harvest. The different diets were administered 2 weeks after the end of the TAM administration cycle (see details below). In the CD switch experiment, mice were fed HFD for 6 months and switched back to CD (HFD-CD) for 4 weeks before organ harvesting. Pomc^CreERT2 mice[23] were generated on a C57BL/6J background and were kindly provided by Joel K Elmquist. The line was crossed with either ROSA-tdTomato (#:007908, Jackson Laboratory, B6;129S6 mixed genetic background) or ROSA-ZsGreen (#:007906, Jackson Laboratory, C57BL/6J background) lineage-tracking mice. Pomc^CreERT2 mice were also crossed with Pomc-eGFP mice (Jackson Laboratory,

#009593, C57BL/6J background) to generate double reporter Pomc^CreERT2;tdTomato;Pomc-eGFP animals. Pomc^Dre mice[17] were back-crossed to a C57BL/6N background and subsequently crossed to homozygous ROSA-ZsGreen mice (ROSA26^rx/rx). Resulting double transgenic Dre + /−ROSA26^rx/wt mice were used as experimental animals. All animals were sacrificed after 4 h of fasting in the light phase, unless otherwise stated. For the fasting-refeeding experiment, one group of CD-fed Pomc^CreERT2;tdTomato mice were either fasted for 24 h or fasted for 24 h and re-fed with HFD 2 h before sacrifice. For in-vivo leptin challenge, CD-fed Pomc^CreERT2;tdTomato mice were fasted for 3 h and pretreated subcutaneously with vehicle (20 mM Tris-HCl, pH = 8.0) or leptin (5 mg/kg in Tris-HCl, Bio-Techne, 498-OB) for 45 min before sacrifice. Transgenic mice were genotyped by PCR and using specific primers (see Table S2 in the supplementary information) from genomic DNA purified from tail biopsies.

### Tamoxifen (TAM) administration
A 1 g/mL suspension of TAM (T4658, Sigma-Aldrich, Saint-Quentin Fallavier, France) was prepared in 95% ethanol (20821.33.0, VWR, Briare, France) and preheated (55 °C) in corn oil (C8267, Sigma-Aldrich, Saint-Quentin Fallavier, France) to obtain a 50 mg/mL stock. Twelve-week-old Pomc^CreERT2;ZsGreen, Pomc^CreERT2;tdTomato and Pomc^CreERT2;tdTomato;Pomc-eGFP mice were treated with 150 mg/kg body weight TAM daily for 5 consecutive days by oral gavage using flexible polypropylene gavage needles (FTP-20, Instechlabs, Plymouth Meeting, PA, USA). The gavage volume was 3 mL/kg body weight. All animals were treated with 1 cycle of TAM injections unless otherwise stated. One group of Pomc^CreERT2;tdTomato mice was treated with 2 cycles of TAM to test the efficiency of TAM-mediated CRE recombination as follows: cycle 1–2 weeks with no treatment-cycle 2.

### Intracerebroventricular surgery and BrdU infusion
Male Pomc^CreERT2;tdTomato;Pomc-eGFP mice aged 16–21 weeks old underwent intracerebroventricular surgery as it follows. Mice were anaesthetised with isoflurane and treated with analgesics (buprenorphine pre-op; meloxicam post-op). A cannula was implanted into the lateral ventricle (coordinates from bregma: AP −0.3 mm; ML + /− 1 mm; DV −2.5 mm) by using a stereotaxic apparatus (Kopf). The cannula was connected to a micro-osmotic pump (Model 1004, Alzet) previously filled with BrdU (5-Bromo-2 −deoxyuridine, Sigma, B9285) diluted in sterile saline with DMSO 22.5% (concentration 4.5 µg/µl; mean pumping rate: 0.11 µL/h; mean fill volume: 97.3 µL). The catheter (Brain infusion kit 2, Alzet) connected to each pump was filled with a sufficient volume of saline to allow 3 days of postoperative recovery before starting the BrdU infusion. For the short-term experiment, one week after surgery, half of the animals were placed on HFD for 3 weeks, while the other half were maintained on chow diet. The animals were then sacrificed 3 weeks after CD vs HFD administration. For the long-term experiment, BrdU was continuously infused for 3 months in animals were pre-fed on CD or HFD for 3 months prior to surgery. The minipumps were

replaced twice to extend the treatment period to a total of 84 days. We used a Y-shaped connector (3-way Y connector, 22ga, Instech Laboratories, SCY22) between the infusion cannula and the pump reservoir to avoid increased intracranial pressure and subsequent tissue irritation. A mixture of DMSO 22.5% in saline was used for vehicle-treated animals. Body composition was measured before surgery to allocate the groups. One week after surgery, half of the animals were placed on HFD for 3 weeks. Body weight and food consumption were recorded at least twice a week for 4 weeks after surgery until the animals were perfused transcardially for neuroanatomical analysis.

## Brain tissue sampling

Mice were deeply anaesthetised with pentobarbital ip and then perfused transcardially with ice-cold PBS, pH 7.4, followed by 4% paraformaldehyde (PFA, Sigma-Aldrich, France). Brains were extracted and post-fixed in 4% PFA overnight at 4 °C, then cryoprotected in 30% sucrose in PBS at 4 °C. Coronal sections (30 μm) were cut using a cryostat (CM1950, Leica, Germany), collected and stored in anti-freeze solution (30% ethylene glycol, 30% glycerol in KPBS) at −20 °C until further use.

## Neuroanatomical analyses

**-Immunohistochemistry (IHC).** Brain sections from Pomc$^{CreERT2}$; ZsGreen, Pomc$^{CreERT2}$;tdTomato and Pomc$^{CreERT2}$;tdTomato;Pomc-eGFP mice were processed as it follows. After three washes in PBS, sections were incubated with the primary antibody in [PBS + 0.3% Triton X-100] overnight at room temperature. The next day, sections were washed three times in PBS and incubated for 2 h with the secondary antibody [PBS + 0.3% Triton X-100]. After three washes in PBS, sections were mounted with ProLong Gold medium, coverslipped (0.13-0.17 mm thickness) and stored in the dark until imaging. For IHC against the marker Pomc, a primary antibody rabbit anti-Pomc (Phoenix Pharmaceuticals, H-029-30, 1:2000) and a secondary antibody donkey anti-rabbit Alexa Fluor 647 (Jackson Immunoresearch, 711-605-152, 1:500) were used. For ZsGreen, a primary antibody guinea-pig anti-ZsGreen (Frontier Institute, Af940, 1:500) and a secondary antibody donkey anti-guinea-pig Alexa Fluor 647 (Jackson Immunoresearch, 706-605-148, 1:500) were used. For tdTomato, we used a primary antibody goat anti-tdTomato (Sicgen, AB8181, 1:500) and a secondary antibody donkey anti-goat Alexa Fluor 647 (Invitrogen, A21447, 1:500). For Tbx3, we used a primary antibody rabbit anti-TBX3 (Bethyl, A303-098A, 1:500) and a secondary antibody donkey anti-rabbit Alexa Fluor 647 (Jackson Immunoresearch, 711-605-152, 1:500). For phospho-Stat3 (Tyr705), sections were preincubated with 100% methanol at −20 °C for 10 min, followed by incubation with rabbit anti-pStat3 primary antibody (Cell Signalling, 9131, 1:500) and donkey anti-rabbit Alexa Fluor 647 secondary antibody (Jackson Immunoresearch, 711-605-152, 1:500). For Ki67, we used a primary antibody mouse anti-Ki-67 (BD Biosciences, 550609, 1:500) and a secondary antibody goat anti-mouse Alexa Fluor 488 (Cell Signalling, 4408 S, 1:500). For Caspase 3, a primary mouse anti-cleaved caspase-3 (Asp175) (Cell Signalling, 9661, 1:500) and a secondary antibody donkey anti-rabbit Alexa Fluor 647 (Jackson Immunoresearch, 711-605-152, 1:500) were used. For BrdU, sections were preincubated with 2 N HCl at 37 °C for 30 min and 0.1 M borate buffer at room temperature for 30 min, followed by incubation with primary rat anti-BrdU (Abcam, ab6326, 1:500) and secondary goat anti-rat biotin (Vector Laboratories, #BA-9401, 1:500). Signal was enhanced with NeutrAvidin-DyLight633 (Invitrogen, #22844, 1:500). For NeuN, a primary chicken anti-NeuN (Millipore, ABN91, 1:2000) and a secondary donkey anti-chicken Alexa Fluor 647 (Jackson Immunoresearch, 703-606-155, 1:500) were used. For Calbindin-D28K, a primary rabbit anti-Calbindin-D28K (Sigma-Aldrich, C2724, 1:500) and a secondary donkey anti-rabbit Alexa Fluor 647 secondary antibody (Jackson Immunoresearch, 711-605-152, 1:500) were used. For Sox2, a primary rabbit anti-Sox2 (EMD Millipore, AB5603, 1:500) and a

secondary donkey anti-rabbit Alexa Fluor 647 secondary antibody (Jackson Immunoresearch, 711-605-152, 1:500) were used.

**- Double fluorescent in situ hybridisation (FISH) combined with IHC.** Fluorescein-labelled riboprobe against mouse *Pomc* (Pomc-FITC, Allen Brain Atlas, #RP_Baylor_102974), digoxigenin-labelled riboprobe against *Cart* (Cart-DIG, Allen Brain Atlas, #RP_071204_02_B05), *Npy* (Npy-DIG, Allen Brain Atlas, #RPS_Baylor_37109). *AgRP* (AgRP-DIG, Allen Brain Atlas, #RP_050419_04_D06), and *Kiss* (Kiss-DIG, Allen Brain Atlas, #RP_080109_03_C10) were prepared from hypothalamic RNA reverse transcribed into cDNA. DNA templates for *Pomc, Cart*, *Npy*, *AgRP* and *Kiss* were obtained via PCR and sub-cloned into pBluescript vector using the restriction enzymes EcoRI and BamHI (*Pomc, Cart, Npy, AgRP*) or EcoRI and SpeI (*Kiss*). After linearization with BamHI, the antisense riboprobes were synthesised with T3 RNA polymerase. For the generation of the sense riboprobes, EcoRI (*Pomc, Cart, Npy, AgRP*) or SpeI (*Kiss*) was used for the linearization and T7 RNA polymerase for the synthesis. 35% of the uridine triphosphate (UTPs) in each reaction was replaced with either DIG-11-UTP for *Cart, Npy, AgRP, Kiss* or FITC-12-UTP for *Pomc*, probe production, respectively. FISH was performed as it follows. Free-floating slices were treated with 0.2 M HCl for 20 min and acetylated with 0.25 % acetic anhydride in 0.1 M triethanolamine (pH = 8.0) for 10 min, before hybridisation of Pomc-FITC and Cart-DIG, AgRP-DIG or Npy-DIG probes, diluted 1:1000 in hybridisation buffer, overnight at 70 °C. On the second day, after several washes with saline-sodium-citrate buffers of increasing stringency at 70 °C, sections were blocked for 30 min in "TNB" blocking buffer (Akoya Biosciences, #FP1012) and incubated for 1 h in sheep anti-DIG-POD antibody (Sigma-Aldrich, #11207733910) diluted 1:1000 in TNB. Cart-DIG, AgRP-DIG, Npy-DIG and Kiss-DIG probes were then detected using Cy3-Tyramide (Akoya Biosciences, NEL744001KT) diluted 1:100 in 1X Plus amplification diluent (Akoya Biosciences, #FP1498) for 10 min. After peroxidases quenching with 3% $H_2O_2$ diluted in PBS for 30 min, sections were incubated for 1 h in sheep anti-FITC-POD (Sigma-Aldrich, #11426346910) diluted 1:1000 in TNB. Pomc-FITC probes were then revealed using Coumarin-Tyramide (Akoya Biosciences, NEL703001KT) diluted 1:100 in 1X Plus amplification diluent (Akoya Biosciences, #FP1498) for 10 min. After peroxidases quenching with 3% $H_2O_2$ diluted in PBS for 30 min, sections were incubated overnight at room temperature in either guinea-pig anti-ZsGreen (Frontier Institute, Af940, 1:500), goat anti-tdTomato (Sicgen, AB8181, 1:500), rabbit anti-TBX3 (Bethyl, A303-098A, 1:500) or rabbit anti-c-Fos (Santa Cruz, sc-52, 1:500). The next day, sections were incubated for 2 h in donkey anti-guinea-pig Alexa Fluor 647 (Jackson Immunoresearch, 706-605-148, 1:500) for ZsGreen revelation, in donkey anti-goat Alexa Fluor 647 (Invitrogen, A21447, 1:500) for tdTomato revelation or donkey anti-rabbit Alexa Fluor 647 (Jackson Immunoresearch, 711-605-152, 1:500) for Tbx3 and c-Fos revelation, in [PBS + 0.3 % Triton X-100]. Finally, sections were rinsed 3 times in 50 mM Tris-HCl (pH = 7.5) and mounted with ProLong Gold antifade reagent (Invitrogen, #P36930).

**-Triple FISH combined with IHC.** Fluorescein-labelled riboprobe against mouse Gad1 (Gad1-FITC, Allen Brain Atlas, #RP_071204_01_D08) and Gad2 (Gad2-FITC, Allen Brain Atlas, #RP_Baylor_253912), digoxigenin-labelled riboprobe against Slc17a6 (Slc17a6-DIG, Allen Brain Atlas, #RP_050921_01_E03) and biotin-labelled riboprobe against Pomc (Pomc-biotin, Allen Brain Atlas, #RP_Baylor_102974) were prepared as mentioned above. For FISH analyses, free-floating slices were treated with 0.2 M HCl for 20 min and acetylated with 0.25 % acetic anhydride in 0.1 M triethanolamine (pH = 8.0) for 10 min, before hybridisation of Gad1-FITC, Gad2-FITC, Slc17a6-DIG and Pomc-biotin probes, diluted 1:1000 in hybridisation buffer, overnight at 70 °C. On the second day, after several washes with saline-sodium-citrate buffers of increasing stringency at 70 °C, sections were blocked for 30 minutes in "TNB" blocking buffer (Akoya Biosciences, #FP1012) and incubated for 1 h in sheep anti-DIG-POD antibody (Sigma-Aldrich, #11207733910)

diluted 1:1000 in TNB. Slc17a6-DIG probes were the detected using Cy3-Tyramide (Akoya Biosciences, NEL744001KT) diluted 1:100 in 1X Plus amplification diluent (Akoya Biosciences, #FP1498) for 10 min. After peroxidases quenching with 3% $H_2O_2$ diluted in PBS for 30 min, sections were incubated overnight at room temperature in sheep anti-FITC-POD (Sigma-Aldrich, #11426346910) diluted 1:1000 in TNB. The next day, Gad1-FITC and Gad2-FITC probes were then detected using Fluorescein-Tyramide (Akoya Biosciences, NEL741001KT) diluted 1:100 in 1X Plus amplification diluent (Akoya Biosciences, #FP1498) for 10 min. After peroxidases quenching with 3% $H_2O_2$ diluted in PBS for 30 min, sections were incubated in Streptavidin-HRP (Akoya Biosciences, TS-000300) diluted 1:100 in TNB for 30 min. Pomc-biotin probes were then revealed using Coumarin-Tyramide (Akoya Biosciences, NEL703001KT) diluted 1:100 in 1X Plus amplification diluent (Akoya Biosciences, #FP1498) for 10 min. After peroxidases quenching with 3% $H_2O_2$ diluted in PBS for 30 min, sections were incubated overnight at room temperature in goat anti-tdTomato (Sicgen, AB8181, 1:500). The next day, sections were incubated for 2 h in donkey anti-goat Alexa Fluor 647 (Invitrogen, A21447, 1:500) in [PBS + 0.3% Triton X-100]. Finally, sections were rinsed 3 times in 50 mM Tris-HCl (pH = 7.5) and mounted with ProLong Gold antifade reagent (Invitrogen, #P36930).

**-Single molecule FISH.** The smFISH technique (RNAscope) was used to detect mRNA of *Pomc*, *Tdtomato*, *Zsgreen*, *Lepr-tv1*, *Glp1r*, *Insr* and *Glipr1*. All reagents were purchased from Advanced Cell Diagnostics (ACD). The *Pomc* probe (314081) contained 10 oligonucleotide pairs targeting region 19-995 (NM_008895.3) of the *Pomc* transcript; the *Tdtomato* probe (317041) contained 20 oligonucleotide pairs targeting region 7-1382 of the *Tdtomato* transcript; the *Zsgreen* probe (461251) contained 15 oligonucleotide pairs targeting region 980-1655 (JQ071441.1) of the *Zsgreen* transcript; the *Lepr-tv1* probe (471171) contained 19 oligonucleotide pairs targeting region 3220-4109 (NM_146146.2) of the *Lepr* transcript variant 1; the *Glp1r* probe (418851) contained 20 oligonucleotide pairs targeting region 108−1203 (NM_021332.2) of the *Glp1r* transcript; the *Insr* probe (482511) contained 20 oligonucleotide pairs targeting region 2000−2920 (NM_010572.2) of the *Insr* transcript; the *Glipr1* probe (1203571) contained 20 oligonucleotide pairs targeting region 113-1060 (NM_028608.3) of the *Glipr1* transcript. RNAscope 4-plex negative (321831) and positive-control probes (321811) were processed in parallel with the target probes. All incubation steps were performed at 40 °C using the ACD HybEz hybridisation system (321711) if not stated otherwise. One day before the assay, sections were mounted on SuperFrost Plus Gold slides (Thermo Scientific, J1800AMNZ), dried at room temperature, briefly rinsed in autoclaved Millipore water, air dried and incubated at 60 °C for 1 h. Subsequently, slides were submerged in Target Retrieval reagent (322000) at 99.5 °C for 10 min, rinsed once in autoclaved Millipore water and dehydrated in 100% ethanol for 3 min. Slides were air dried for 1 h at 60 °C, a hydrophobic barrier was created around the sections using an ImmEdge hydrophobic barrier pen (Vector Laboratories, H-4000) and slides were stored at room temperature until assaying. The following day, slides were incubated with Protease Plus for 30 min. The subsequent steps, that is, hybridization of the probes, amplification and detection steps, were performed according to the manufacturer's protocol for RNAscope Fluorescent Multiplex Detection Reagent kit (323110), or for more than three probes, the protocol for RNAscope 4-Plex Ancillary Kit for Multiplex Fluorescent Kit (323120) was used. The probes were detected using tyramide-diluted Opal690 (1:1500), Opal650 (1:750), Opal620 (1:1500), Opal570 (1:1500) or Opal520 (1:750). Sections were counterstained with DAPI and coverslipped with ProLong Gold Antifade Mountant (Thermo Fisher, P36930), cover-slipped (0.13−0.17 mm thickness) and stored in the dark until imaging.

**CUBIC.** Mice were deeply anaesthetised with pentobarbital ip and then perfused transcardially with ice-cold PBS, pH 7.4, followed by 4% paraformaldehyde (PFA, Sigma-Aldrich, France). Brains were extracted and post-fixed in 4% PFA overnight at 4 °C. The next day, brains were washed 3 times 2 h in PBS. For delipidation, brains were first incubated in 50% CUBIC-L reagent (T3740, Tokyo Chemical Industry, Japan) diluted in water for 24 h at 37 °C and then in 100% CUBIC-L reagent for 5 days. CUBIC-L solution was refreshed on days 2 and 4. Brains were then washed 3 times 2 h in PBS at RT. The following step consisted in refractive index matching and started with 50% CUBIC-R (T3741, Tokyo Chemical Industry, Japan) diluted in water for 24 h at RT followed by 100% CUBIC-R for 24 h at RT. CUBIC cleared brains were acquired on a Ultramicroscope II light sheet microscope (LaVision Biotec, Germany) using a sCMOS camera (Andor Neo) and a 2x/0.5 objective lens.

### Image acquisition and data analysis

The processed single-cell RNAseq data (HypoMap) is available online (https://www.repository.cam.ac.uk/handle/1810/340518). The R package Seurat (version 4.3.0) was used to subset the data and generate scatter- and dotplots. The R package schex (version 1.12.0) was used to generate the hexplots used in the schematic view in Fig. 1 that correspond to the single cell scatterplots in the extended data figures. In each hexagonal bin the gene expression of all cells is averaged. Outlier cells were removed by removing hexagonal bins with fewer than 10 cells. IHC, FISH and smFISH images were acquired using Leica Application Suite (v3.5.7) and a Leica DM6 CFS TCS SP8 confocal microscope equipped with an X20/1 dry objective available at the Bordeaux Imaging Centre (BIC). Z-stacks were acquired with optical sections of 0.9 µm. CUBIC optically cleared brains were acquired on a Ultramicroscope II light sheet microscope (LaVision Biotec, Germany) using a sCMOS camera (Andor Neo) and a 2x/0.5 objective lens. The objective was equipped with a 6 mm working distance dipping cap. Zoom was set to 2x and Z-step size was 5 µm so the voxel sizes were 1.625 × 1.625 × 5 µm. The eGFP samples were excited with a 488 nm laser and the tdTomato with a 561 nm laser intensities were kept constant under all relevant conditions. Images were imported into FIJI (National Institute of Health, version 1.53t). For representative images, brightness and contrast adjustments were kept constant for each channel across all related conditions. For quantification, all channels were left unchanged and an average of 2 medial ARC sections were quantified per mouse (4 hemi-ARC). For FISH and IHC quantification, maximum intensities were projected and reporter positive cells were segregated using visual threshold and the Analyze Particle plugin to define single-cell ROIs. Mean fluorescence intensities for every channel were then measured in each reporter positive cells and positivity for each marker was assessed by individual thresholds. For smFISH and CUBIC quantification, reporter positive cells were segregated in 3D using the 3D Suite plugin to define single-cell 3D ROIs. Mean and maximum fluorescence intensities for every channel were then measured in each reporter positive cells. Mean intensities for *Pomc* were used to define Pomc+ vs Ghost cells. Maximum intensities for *Glipr1*, *Lepr*, *Glp1r* and *Insr* were used to define positivity for the marker. For CUBIC quantification, X, Y and Z spatial positions of every single-cell ROIs were also extracted. A Python script was used to generate isosurface density plots. The probability density function of the 3D distribution of neurons was estimated for the entire traced cell population, as well as for the Pomc+ and Ghost subpopulations. This was performed using the non-parametric kernel density estimation method[17]. The density of neurons within the ARC was thus calculated based on the 3D neuronal coordinates to demonstrate the spatial concentration, which was then colour coded to visualise the regions within the ARC with higher neuronal counts for each subpopulation.

## Electrophysiology and transcriptomic analyses

**Brain slice preparation and recordings.** Mice were euthanatised (Exagon/lidocaine: 300/30 mg/kg, i.p.) and perfused intracardially with ice-cold NMDG solution containing (in mM): 1.25 $NaH_2PO_4$, 2. 5 KCl, 7 $MgCl_2$, 20 HEPES, 0.5 $CaCl_2$, 28 $NaHCO_3$, 8 D-glucose, 5 L(+)-ascorbate, 3 Na-pyruvate, 2 thiourea, 93 NMDG and 93 HCl 37%; pH: 7.3−7.4; osmolarity: 305−310 mOsM. Brains were quickly removed and placed in an oxygenated resting solution of NMDG at 34 °C for 10 min. Then, 250 µm slices containing the DR were cut with a vibroslice (Leica VT1000S, Wetzlar, Germany) and transferred at room temperature to aCSF solution (containing the following in mM: 124 NaCl, 2.5 KCl, 1.25 $NaH_2PO_4$, 2 $MgCl_2$, 2.5 $CaCl_2$, 2.5 D-glucose and 25 $NaHCO3$; pH: 7.3−7.4; osmolarity: 305−310 mOsM) for at least 30 min. For whole-cell recordings, borosilicate pipettes (4−6 MΩ; 1.5 mm OD, Sutter Instrument) were filled with an intracellular K-gluconate solution (containing the following (in mM): 128 K-gluconate, 20 NaCl, 1 $MgCl_2$, 1 EGTA, 0.3 $CaCl_2$, 2 $Na_2$-ATP, 0.3 Na-GTP, 0.2 cAMP and 10 HEPES; pH: 7.3−7.4; osmolarity: 280−290 mOsM). For patch-qPCR experiments, 80 U RNAse inhibitor RiboLock (Thermo Scientific, EO0382) was added to the intracellular solution. Recordings were made using a Multiclamp 700B amplifier, digitised using a Digidata 1440 A interface and acquired at 2 kHz using pClamp 10.5 software (Axon Instruments, Molecular Devices, San José, CA, USA). Pipette and cell capacitances were fully compensated and junction potential corrected off-line. Electrophysiological data were collected using Clampex (Molecular Devices, v11.1).

**Evaluation of leptin and insulin responsiveness of Pomc neurons.** Using Pomc^CreERT2;tdTomato;Pomc-eGFP mice, we detected Pomc+ and Ghost neurons based on tdTomato and Pomc-eGFP expression using an electrophysiology setup coupled to an epifluorescence microscopy system (pE-2 CoolLED excitation system, UK) and investigated dynamic electrophysiological properties. Neuronal electrophysiological properties were monitored in a whole-cell configuration using the protocol described in the previous section from ARC slices perfused with oxygenated aCSF. For each recorded neuron, membrane potential (Em) and membrane capacitance (Cm) were recorded in voltage-clamp configuration (Vh = −60 mV) immediately after cell opening. Input resistance (Ri) and action potential (AP) properties were measured by injecting current steps in the current-clamp configuration (start: −60 pA; Δ+10 pA to rheobase). Action potential properties were assessed at rheobase. Responses to 5 min application of leptin (200 nM, Bio-Techne, 498-OB) or insulin (150 nM, Sigma-Aldrich, I031000) were monitored after 10 min baseline. Membrane potential and input resistance were monitored during the whole experiment in response to constant hyperpolarizing pulse (−5 pA). Leptin- and insulin-induced changes in membrane potential were assessed by measuring the average membrane potential for 30 s, before and 5 min after the application hormonal application. Input resistance was calculated in response to hyperpolarizing 10 pA currents from 0 to −80 pA. Neurons with at least 2 mV change in the membrane potential and 10% change in the input resistance after leptin or insulin applicSation were considered as responsive neurons.

**Single-cell Patch-seq.** Neuronal intrinsic properties, action potential properties, and EPSCs properties were analyzed as described above in Pomc^CreERT2;tdTomato mice fed with CD or HFD for 6 months. At the end of the recording, each cell in the pipette was aspirated for 3 min. The pipette was then broken into an RNAse-free 200 µL Eppendorf tube containing the reaction mix from the SMART-Seq® Single Cell Kit (Takara, 634472). cDNA synthesis was performed according to the manufacturer indications using the mastercycler® nexus gradient (Eppendorf, Germany). Once the cDNAs had been amplified, they were frozen and transferred to the GeT-Santé facility (Inserm, Toulouse, France, get.genotoul.fr), where they were purified according to the Takara Bio guidelines, and checked on a Fragment Analyzer run,

(Agilent Technologies, Santa Clara, USA), using the High Sensitivity NGS Kit. Libraries have been prepared with the Nextera XT DNA Library Prep Kit (Illumina, San Diego, USA), according to Illumina's protocol with some adjustments. Briefly, between 353 and 1000 pg of purified cDNA were first tagmented (fragmentation and tagging with adapters at the same time). Then, compatible unique dual indexed primers were added and 12 cycles of PCR were applied to amplify libraries. Libraries were purified with AMPure XP beads using a two-sides strategy to obtain 280−700 pb fragments. Libraries quality was assessed using the HS NGS kit on the Fragment Analyzer (Agilent Technologies, Santa Clara, USA) and quantified by Qubit and the dsDNA HS Assay Kit (Thermofisher Scientific, Waltham, MA, USA). Libraries were equimolarly pooled and sequenced at the GeT-PlaGe core facility (INRAE, Toulouse, France), on one SP lane of the Illumina NovaSeq 6000 instrument (Illumina, San Diego, USA), using the NovaSeq 6000 SP Reagent Kit v1.5 (300 cycles), and a paired-end 2 × 150 pb strategy. Between 2.3 and 10.7 million raw paired-end reads were produced per library.

**Validation Pomc expression in Ghost cells by single-cell qPCR.** After obtaining ex-vivo living brain slice from Pomc^CreERT2;tdTomato; Pomc-eGFP mice, we have aspirated cytosolic cell content form 5 different Ghost cells using borosilicate pipettes. Target cells were identified on the basis of positivity for the tdTomato signal and the absence of eGFP endogenous expression. We used a single cell qPCR protocol to amplify *Pomc* mRNA and the housekeeping gene Eef1a1. The pipette was broken into an RNAse-free 200 µL Eppendorf tube containing the reaction mix from the Maxima First Strand cDNA Synthesis Kit (Thermo Scientific) according to the manufacturer's instructions and supplemented with 2% TritonTM X-100 (Sigma-Aldrich, T8787). After harvesting, the retrotranscription of single cell mRNA was performed using the mastercycler® nexus gradient (Eppendorf, Germany). All cDNA was generated after increasing temperature steps as follows: 25 °C for 10 min, 50 °C for 60 min and 70 °C for 15 min. Nested pre-amplification PCR was then performed on the LightCycler® 480 (Roche Applied Science, Manheim, Germany) using PerfeCTa PreAmp SuperMix (Quanta Biosciences) with template cDNA and primers mix at 0.5 µM (Table S1). The reaction mixture was subjected to 17 PCR cycles consisting of an initial denaturation step at 95 °C for 2 min followed by 17 cycles of 95 °C for 15 s, and 60 °C for 3 min. Unincorporated primers were removed by exonuclease digestion (NEB, Ipswich, MA, USA). The second amplification was performed using LightCycler® 480 SYBR Green I Master (Roche Applied Science) on amplified cDNA in a reaction of 10 µl, using the nested primers at 0.6 µM. An initial denaturation step at 95 °C for 5 min was followed by 45 cycles of 95 °C for 15 s, and 60 °C for 30 s. PCR products were run on an automated electrophoretic DNA separation system (Labchip GX II, Caliper life sciences, MA, USA). For each primer pair a no-template-control without cDNA was also tested, and no PCR products were observed. The primers used are listed in Table S2. Positive (hypothalamic homogenate) and negative control [VMH (ventromedial hypothalamus)] samples were included, as indicated (Table S1). Additional negative control samples were used to exclude potential mRNA contamination in the intrapipette solution or the reaction mix used to collect the cytoplasmic content (Table S1). The PCR products were collected using the Mix2SeqKit (Eurofins), and DNA sequencing (Sanger) was performed by Eurofins. After blasting both the forward and reverse sequences obtained (Supplementary file 1), we have successfully verified 100% homology with the mouse *Pomc* gene, thus validating the presence of specific *Pomc* mRNA expression in Ghost neurons.

**Data collection and statistical analyses.** Cluster analysis of and single-cell sequencing data was performed using an in-house Python (v3.3) script. Based on the normalised gene expression for each single cell, Leiden algorithm was used in combination with the Silhouette test

to define the optimal number of clusters. All other statistical analyses were performed using Prism (v9.5, Graphpad, USA). We used FIJI (National Institute of Health, v1.53t) for all image analysis. Electro-physiological data were analyzed using Esay Electrophysiology (v2.5.2). All values are expressed as mean ± SEM unless otherwise noted. Data were analyzed using unpaired or paired Student's t-tests or 1-way or 2-way ANOVAs, as appropriate. Normality of the variable was test with a Shapiro-Wilk test before performing ANOVAs. Significant ANOVAs were followed by Tukey's post-hoc test. Chi-squared test was used to describe changes in the percentage of cells responding or not responding to hormones (Fig. 3E, F) and changes in the percentage of cells in each cluster (Fig. 4H). $P < 0.05$ indicates statistical significance.

### Reporting summary
Further information on research design is available in the Nature Portfolio Reporting Summary linked to this article.

## Data availability
Source data and representative images are provided with this paper. All data supporting the results of this study are available in the paper and supplementary information files. Additional original data supporting the findings are available on request from the corresponding author. All materials, data, code and associated protocols will be made available to readers without undue restriction. The processed single-cell RNA-seq data analysed in Fig. 1 and Fig. S2 (HypoMap) are available online (https://www.repository.cam.ac.uk/handle/1810/340518). The scRNA-seq data generated in this study have been deposited in the Gene Expression Omnibus (GEO) database under the accession code GSE261715. Further information on research design is available in the Reporting Summary linked to this paper. Source data are provided with this paper.

## Code availability
Python code used for isodensity plots generation and 3D statistical analysis was provided by Jens Brüning and is available at https://github.com/bruening-lab/Heterogeneity_Scripts. Python code used for cluster analysis of scqPCR and scRNA seq analyses was provided by Yves Le Feuvre and is available upon request.

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

## Acknowledgements

C.Q. acknowledges INSERM, Agence Nationale Recherche (ANR-20-CE14-0046), French Societies of Endocrinology, Nutrition and Diabetes (SFE, SFN and SFD), Fyssen Foundation and Institut Benjamin Delessert. D.C. thanks INSERM, Nouvelle Aquitaine Region, IdEx University of Bordeaux/GPR BRAIN_2030, Labex BRAIN ANR-10-LABX-43, ANR-18-CE14-0029, ANR-21-CE14-0018, ANR-22-CE14-0016, and Fondation Recherche Médicale (FRM-EQU202303016291). S.L. and C.A. acknowledge the Fondation Recherche Médicale (FDT202204014771 for S.L. and FRM-ARF201809006962 for C.A.). Microscopy was performed at the Bordeaux Imaging Centre (BIC), a service unit of the CNRS-INSERM and the University of Bordeaux, member of the national infrastructure France BioImaging, supported by Labex Brain. The authors thank F. Cordelières, C. Poujol, M. Mondin, S. Marais and J. Teillon (BIC, University of Bordeaux). We thank Fabrice Cordelières (BIC) for assistance with image analysis and macro generation. We thank the animal housing and genotyping facilities of the INSERM U1215 Neurocentre Magendie, Elisabeth Huc, Fiona Corailler, Jean-Baptiste Bernard and Ruby Racunica (INSERM U1215) for their exceptional help with animal breeding and housing. We thank Jeflie Tournezy for help with c-FOS analysis. The authors thank the GeT-Santé facility (I2MC, Inserm, Génome et Transcriptome, GenoToul, Toulouse, France) for the advice and technical contribution to the experiments.

## Author contributions

C.Q. conceived and designed the study and wrote the manuscript, with contributions from D.C. and S.L. S.L. performed the majority of the experiments and analyzed the data with contributions from C.Q. V.S., T.H.L., L.S., S.J., N.B., T.L.L., N.D., A.C., L.B., P.Z., C.A., D.G., Y.L.F., E.L., A.B., J.T., J.C.N. and M.N. contributed to the experiments and data analysis. G.M., X.F., J.C.B. and D.C. contributed to the discussion and provided vital material.

## Competing interests

The authors declare no competing interests.
