## [Peer Review File · Nature Communications]

Single cell tracing of Pomc neurons reveals recruitment of 'Ghost' subtypes with atypical identity in a mouse model of obesityEditorial Note: This manuscript has been previously reviewed at another journal that is not operating a transparent peer review scheme. This document only contains reviewer comments and rebuttal letters for versions considered at *Nature Communications* .

REVIEWERS' COMMENTS

Reviewer #1 (Remarks to the Author):

I appreciated the effort of the authors have made to argue for a new type of ghost cells from POMC neurons in response to HFD. However, I am sorry to say that I am still not convinced that the evidence as presented convincingly demonstrate a formation of a functionally distinct group of non-POMC neurons from POMC cell lineage that contribute to DIO. The evidence indeed suggests an existence of a subset of POMC neurons that respond to HDF by altering levels of a series of genes including reducing POMC expression. However, it is unknown whether these changes contribute to DIO or represent a secondary effect to HFD.

There is no mechanism identified for the proposed formation of new type of neurons nor a functional contribution of these proposed ghost cells to DIO was shown. Thus, in my review the results illustrated do not support the main claim here.

Reviewer #3 (Remarks to the Author):

The authors have addressed my comments raised during the second revision of this manuscript. I appreciate their efforts in incorporating these new insights and adjusting the manuscript in accordance with my suggestions.